# HMFusion: Hierarchical Multi-Modality Fusion for CAD Representation

## Abstract

Computer-Aided Design (CAD) generation, which plays a vital role in product iteration and virtual simulation, has been of great interest in modern industry. Existing deep learning-based methods for CAD generation have achieved remarkable success. However, these studies either require lengthy domain-specific prompts or multiview sketches. Though effective, they exhibit limitations in ensuring consistency in geometric representation and rely on multiple inputs. To address these challenges, we propose HMFusion: Hierarchical Multi-Modality Fusion for CAD Representation, which incorporates a cross-modal geometric prior with hierarchical embeddings for consistent and faithful CAD generation. Specifically, our method introduces a prompt-enhancement module that transforms minimal user prompts into professional CAD-oriented descriptions containing structural and dimensional details. To improve the consistency of geometric representations, we tightly fuse textual and geometric information through a CAD-aware hierarchical alignment between visual and textual semantics in a hyperbolic space. Extensive experiments demonstrate that our proposed framework achieves effective geometric accuracy and semantic fidelity.

## 1 Introduction

Computer-Aided Design (CAD) is essential for product iteration, interactive visualization, and virtual simulation in modern industry and digital content creation, due to its precise geometric representations and reproducibility (Heidari & Iosifidis, 2024). By encoding geometry as parametric boundary representations built from ordered 2D sketches and 3D operations, CAD systems provide micrometer precision and a complete, traceable record of each design step. As demand for automation and custom design continues to grow, the ability to quickly create high-quality CAD models is becoming a central driver of innovation in design workflows (Regenwetter et al., 2022).

Existing studies primarily focus on two paradigms: Text-to-CAD and Image-to-CAD. The former maps natural language prompts to 3D geometric representations using large language models (LLMs) to interpret detailed textual descriptions. These descriptions are subsequently decoded into structured CAD commands via sequence-to-sequence geometric decoders (Khan et al., 2024b). The latter conditions contextual visual inputs such as multi-

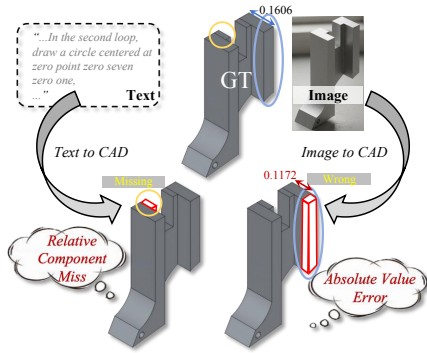

Figure 1: Our motivation: for users, text-only CAD generation demands exhaustive domain-specific detail, and any omission yields missing components; image input captures shape but lacks absolute dimensions, resulting in inaccurate models.

view sketches or point cloud data, and utilizes convolutional or Transformer-based architectures to directly reconstruct 3D (Li et al., 2025c; Chen et al., 2025a) meshes or progressively parameterize sketch elements (Alam & Ahmed, 2024; Wang et al., 2025b; Chen et al., 2024). Both paradigms have achieved notable progress, highlighting the great potential of generative design procedures.

Figure 2: Designers can efficiently generate CAD models from a concise text prompt and a rough sketch or a real photo. Our framework CAD-awarely aligns and fuses these two modalities in a hyperbolic space.

Though effective, as shown in Figure 1, the aforementioned approaches face two major challenges that hinder their broader applicability in interactive design and rapid prototyping: 1) **Reliance on textual descriptions**, particularly in Text-to-CAD methods, which often require lengthy and domain-specific prompts to capture part-level geometry, assembly relations, and dimensional tolerances (Khan et al., 2024b). Without such detailed textual descriptions, these models are apt to semantic-geometry inconsistencies, reducing the accuracy and reliability of the generated CAD outputs. 2) **Lack of consistency in geometric representation**, as existing methods struggle to model absolute and relative geometry simultaneously. Text-to-CAD methods tend to specify absolute dimensions but often fail to capture precise spatial relations between components. Conversely, Image-to-CAD methods excel at relative positioning yet lack accuracy in defining explicit geometric parameters. This inconsistency leads to incomplete or imprecise CAD outputs.

To bridge these gaps, we turn to a multimodal strategy that combines the complementary strengths of language and geometry. This choice raises two key questions. First, how can a terse user prompt be expanded into a CAD-oriented description without imposing additional effort on non-expert users? Second, how can the enriched text and a sketch contour be fused in a representation space so that they can cooperate when guiding the CAD command decoder?

Accordingly, we propose HMFusion: Hierarchical Multi-Modality Fusion for CAD Representation, which is a framework capable of generating high-precision CAD models from only a brief text prompt and a single sketch contour (or a real-world photo), as shown in Figure 2. Our core idea lies in robust CAD model generation through cross-modality information alignment and fusion. First, users provide a simple contour sketch representing the desired CAD shape. A LoRA-tuned large language model (LLM) (Hu et al., 2022) then automatically expands a brief natural-language prompt into a detailed, CAD-oriented description. This prompt-enhancement stage eliminates the traditional dependence on exhaustive, expert-level text by distilling domain knowledge into the LLM itself. Next, we perform CAD-aware hierarchical alignment and fusion between visual and textual semantics in a hyperbolic space (Pal et al., 2024). Cross-modal contrastive learning in this space guides a Transformer-based decoder (Vaswani et al., 2017) to generate a semantically consistent and structurally complete sequence of CAD instructions.

Extensive experiments demonstrate that even with minimal input consisting of brief sentences and an image, the proposed framework achieves superior geometric accuracy and semantic fidelity compared to existing methods, validating its effectiveness in practical CAD generation scenarios. Our contributions can be summarized as follows:

- We propose a multimodal generation paradigm that combines an expert-tuned prompt-enhancement module, implemented with LoRA fine-tuning on a large language model, and a contour-based geometric prior. This pairing converts a brief user prompt into a detailed CAD-oriented description, aligns language with shape information, and jointly alleviates the limitations of text-only or image-only conditioning.

- We introduce a CAD-aware hierarchical embedding equipped with multi-granularity contrastive losses in hyperbolic space, which tightly aligns and fuses textual and visual features across scales. The resulting representation improves structural consistency and supports faithful, robust CAD generation.

## 2 RELATED WORK

### 2.1 CAD GENERATION

Most CAD generation research focuses on generation based on complete 3D information, such as point clouds (Ma et al., 2024; Wu et al., 2021; Khan et al., 2024a), sketches (Li et al., 2020; Wang et al., 2024; Karadeniz et al., 2024), B-reps (Willis et al., 2021; Xu et al., 2021; Li et al., 2025b), and voxel grids (Li et al., 2023; 2024). Some are close to ours, DeepCAD (Wu et al., 2021) pioneered the sketch-extrusion-based construction sequence for CAD modeling, reconstructing a CAD model from latent vectors and point clouds. Recently, Text2CAD (Khan et al., 2024b) proposed multi-level textual inputs ranging from beginner to expert to generate parametric CAD models. CADCrafter (Chen et al., 2025a) is able to generate CAD sequences with synthetic and real-world images. CAD-GPT (Wang et al., 2025a) expanded input modalities by accepting image-based or text-based inputs. While CAD-GPT enabled cross-modal compatibility in CAD creation, it omitted vision-language feature fusion, leaving visual and textual representations as parallel but disconnected features. More recently, several works have explored LLM-based CAD generation with various emphases: CAD-MLLM (Xu et al., 2024) focuses on multimodal conditions, CAD-Llama (Li et al., 2025a) on text-to-CAD based on LLM, CAD-Coder (Guan et al., 2025) on chain-of-thought reasoning, and CADmium (Govindarajan et al., 2025) on code-specific fine-tuning for sequential design.

### 2.2 LEARNING IN HYPERBOLIC SPACE

Building a hierarchical structure or a tree-like structure of data benefits representation learning. Since the Euclidean space has been proven to be unable to encode comparably low distortion for trees by Bourgain's theorem (Linial et al., 1994), hyperbolic space (Nickel & Kiela, 2017; Chamberlain et al., 2017) becomes an alternative to address this issue. The intrinsic hierarchical structure can be maintained with little distortion by generating embeddings in hyperbolic space from such data. This led to the use of hyperbolic models in various modalities, such as text (Tifrea et al., 2018), graph (Franco et al., 2023), and visual data (Long et al., 2020; Franco et al., 2023; Atigh et al., 2022). More recently, HyCoCLIP (Pal et al., 2024) combines hierarchical text and image to learn multimodal representations in hyperbolic space to benefit its hierarchical bias. Our work employs hyperbolic space to learn CAD construction details from both CAD descriptions and images.

## 3 METHOD

Conventional CAD generation pipelines demand either lengthy, step-wise textual programmes or rich geometric scaffolds, placing a high burden on end-users. We propose a CAD-aware multi-modal framework that reconstructs accurate models from a brief natural language description and a rudimentary 2D sketch, as Figure 3. The method first trains a Transformer autoencoder to embed and faithfully recover CAD command sequences, yielding a latent vector $\mathbf{z}$. A domain-adapted Qwen-2.5 model (Yang et al., 2024) then expands the brief user prompt to an expert-level specification, while a contour extractor provides a geometric prior. Textual and visual features are mapped into a shared hyperbolic space and aligned through hierarchical contrastive and entailment objectives tailored to CAD primitives. The fused feature guides the pre-trained decoder, producing a complete command sequence that is deterministically rendered as a boundary representation or mesh, thereby enabling efficient CAD reconstruction from minimal input.

### 3.1 CAD SEQUENCE RECONSTRUCTION

The vocabulary is restricted to the two command families most frequently encountered in industrial practice, sketch and extrusion, following DeepCAD (Wu et al., 2021). In sketch mode, a loop begins with $\langle \text{SOL} \rangle$ and is composed of line (L), arc (A), or circle (R) primitives; extrusion commands subsequently lift the 2D profile into 3D or apply Boolean operations (new body, union, subtraction). All continuous and discrete parameters are uniformly quantized to 256 levels (8-bit) as in (Wu et al., 2021; Chen et al., 2025a). Each instruction is formulated as:

$$C_i = (t_i, p_i), \quad p_i = [x, y, \alpha, f, r, \theta, \phi, \gamma, p_x, p_y, p_z, s, e_1, e_2, b, u], \tag{1}$$

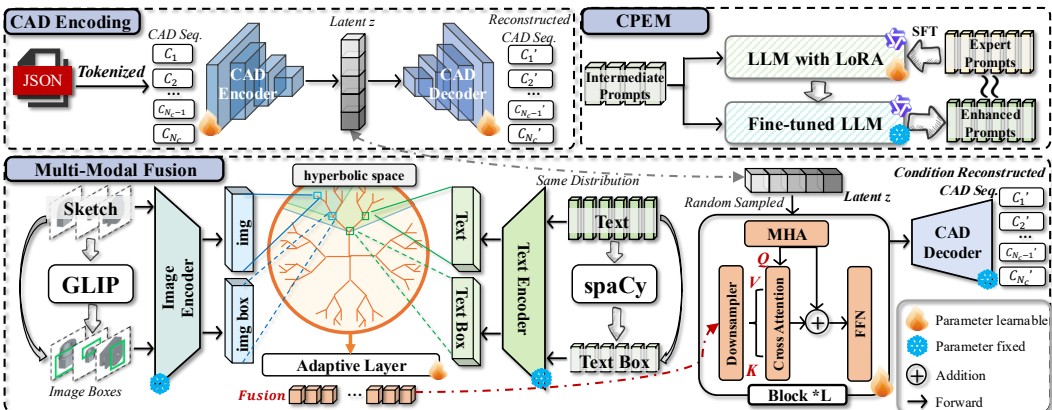

Figure 3: Our pipeline: CAD-sequence auto-encoder is trained to accurately reconstruct models from latent features. Next, a LLM is fine-tuned to perform CAD-oriented textual expansion. Finally, our multi-modality fusion module CAD-aware aligns and fuses the expanded text and contour features in hyperbolic space, guiding the sampled latent representation to generate a CAD model.

and its embedding is obtained by summing the type, parameter, and positional vectors as:

$$e(C_i) = e_i^{\text{cmd}} + e_i^{\text{para}} + e_i^{\text{pos}} \in \mathbb{R}^{256}. \tag{2}$$

Unused parameters are set to $-1$, sequences are padded to $N_c$ with $\langle \text{EOS} \rangle$. The embedded sequence is processed by 4 Transformer layers and average-pooled to yield the latent vector $\mathbf{z} \in \mathbb{R}^{256}$.

The decoder comprises 4 Transformer layers. It is driven by a trainable query sequence, while its keys and values originate from the latent code $\mathbf{z}$ produced by the encoder. During auto-encoder pre-training, $\mathbf{z}$ is computed purely from the input CAD programme. In the later generative phase, the decoder uses the same $\mathbf{z}$ while cross-attention blocks are also used to receive the text- and image-derived embeddings, thereby injecting multimodal guidance. Reconstruction employs (i) cross-entropy loss on command types and (ii) an $\ell_2$ regularization on parameters, replaced by the Huber metric for outlier robustness. The two terms are balanced by a fixed weight $\beta = 2$.

## 3.2 CAD GENERATION FROM MULTI-MODALITY CONDITIONS

**CAD Prompt Enhancement Module (CPEM)**: We begin with a brief description and use expert-level CAD descriptions as supervision to fine-tune a Qwen-2.5 language model, enabling it to understand CAD-related concepts, as shown in Figure 3 (CPEM). This process enhances the model's ability to generate accurate CAD outputs from concise prompts (Vatsal & Dubey, 2024). Additionally, a two-dimensional image serves as a compact geometric prior to guide the shape generation.

The enhanced prompt and image are first embedded and mapped into a common hyperbolic latent manifold, ensuring comparable curvature across modalities. Cross-attention layers then inject this bimodal context into the latent vector $\mathbf{z}$ (as defined in §3.1), so that textual semantics and silhouette geometry jointly guide the generative path. During inference, $\mathbf{z}$ is sampled from the learned latent distribution with the latent GAN technique (Wu et al., 2020), which is also used by DeepCAD (Wu et al., 2021), providing stochastic yet structure-aware variation. The fused representation guides the pretrained auto-decoder, which emits the complete CAD command sequence. A deterministic boundary representation or mesh conversion finally materialises the three-dimensional model.

## 3.3 CAD-AWARE MULTI-MODALITY FUSION

Text-only prompts often lack geometric exactitude; Dimensions, incidence angles, and Boolean semantics are routinely omitted or ambiguous (Picard et al., 2024). In contrast, a single image suppresses key functional cues (for example, hidden edges and internal voids) (Yin et al., 2020). Each modality in isolation yields an incomplete and occasionally misleading specification, which leads to topological errors and unreliable parameter inference during CAD sequence generation.

Therefore, we propose to fuse images and texts by mapping them into a hyperbolic learning space. Given $K$ paired samples $(T_k, I_k)_{k=1}^{K}$, each image $I_k$ is decomposed into image-boxes $I_k^{\text{box}}$ that capture local primitives such as lines, arcs, and profiles by means of GLIP (Li et al., 2022). In parallel, every textual description $T_k$ is segmented into syntactic text-boxes $T_k^{\text{box}}$ using spaCy (Honnibal et al., 2020). This fine-grained partition reflects the hierarchical nature of CAD construction and helps the network learn explicit correspondences between linguistic tokens and geometric components. We aggregate the two objectives through a weighted sum to obtain the overall CAD-aware hierarchical learning as:$\mathcal{L}_{\text{Ch}} = \mathcal{L}_{\text{Chc}} + \gamma\,\mathcal{L}_{\text{Che}}$,where $\gamma$ is a fixed scalar hyper-parameter.

The next subsections present the two components in detail: CAD-Aware Hierarchical Contrastive Learning ($\mathcal{L}_{\text{Chc}}$) and CAD-Aware Hierarchical Entailment Learning ($\mathcal{L}_{\text{Che}}$). The aligned features are then fused and guide the autodecoder(§3.1) step by step, allowing it to assemble complete CAD command sequences with improved structural fidelity.

### 3.3.1 CAD-AWARE HIERARCHICAL CONTRASTIVE LEARNING

Unlike natural images, a CAD drawing is a nested composition of geometric primitives (lines, arcs, circles) that form loops, profiles, and finally volumetric operations. A contrastive objective that ignores this hierarchy tends to conflate unrelated parts and produce weak geometric supervision. We therefore embed both modalities in a hyperbolic space whose negative curvature naturally accommodates the tree-like structure of CAD assemblies.

Let $f_M(\cdot)$ ,$M = \{I, T\}$ denotes encoders for text and image inputs. Let $g_M(M_k) = exp_0^{\kappa}(f_M(M_k))$ ,$M = \{I, T\}$ denotes the hyperbolic representation of text and image. The contrastive loss over pair-level CAD in a batch $B$ is formulated as:

$$\mathcal{L}_{pair}(I, T) = -\sum_{i \in B} \log \frac{\exp\big(d_{\mathcal{L}}(g_I(I_i), g_T(T_i))/\tau\big)}{\sum_{k=1, k \neq i}^{B} \exp\big(d_{\mathcal{L}}(g_I(I_i), g_T(T_k))/\tau\big)}, \tag{3}$$

where the negative Lorentzian distance is taken as a similarity metric and formulated with the softmax, using temperature $\tau$, and negatives for an image are picked from the texts in the batch.

To respect the primitive-level granularity of CAD construction, each image $I_i$ is decomposed into *image-boxes* $\{I_i^{\text{box}}\}$ that isolate individual primitives via GLIP (Li et al., 2022), while each prompt $T_i$ is segmented into *text-boxes* $\{T_i^{\text{box}}\}$ using spaCy (Honnibal et al., 2020). Box-wise alignment prevents the model from matching an entire description to an unrelated fragment of the drawing. The primitive-level CAD contrastive loss is obtained by contrasting every box only against full samples from the remainder of the batch, as $\mathcal{L}_{\text{pair}}(I^{\text{box}}, T)$ and $\mathcal{L}_{\text{pair}}(T^{\text{box}}, I)$.

The final CAD-aware hierarchical contrastive learning $\mathcal{L}_{\text{Che}}$ is formulated as:

$$\mathcal{L}_{\text{Chc}}(I, T, I_{box}, T_{box}) = \frac{1}{4}\Big(\mathcal{L}_{pair}(I, T) + \mathcal{L}_{pair}(T, I) + \mathcal{L}_{pair}(I^{box}, T) + \mathcal{L}_{pair}(T^{box}, I)\Big). \tag{4}$$

### 3.3.2 CAD-AWARE HIERARCHICAL ENTAILMENT LEARNING

CAD model contains one or multiple primitives such as cylinder, rectangular. These components also exist in its image and text description, appearing in the form of a local image and a noun or phrase. Considering the local-global relationship in image and text, we use hierarchical Compositional Entailment to learn the relationship between primitives and entire models. Entailment cones (Ganea et al., 2018) define a region $\mathcal{R}_q$ for every point $q$, all points $p \in \mathcal{R}_q$ are linked to $q$ as its child concept. Points in $\mathcal{R}_q$, referring to the entire image or text, contain more specific information compared to point $q$, which refers to the information at the box-level. Considering the Lorentz model, the half-aperture of these conical regions is formulated by (Le et al., 2019; Desai et al., 2023):

$$\omega(q) = \sin^{-1}\Big(\frac{2K}{\sqrt{\kappa}\,\|\tilde{q}\|}\Big), \tag{5}$$

where $-\kappa$ is the curvature of the hyperbolic space and $K = 0.1$ is a constant to limit values near the origin (Ganea et al., 2018). Based on the prerequisite that the aperture varies inversely with the norm $\|\tilde{q}\|$, a general concept with a wider aperture would lie closer to the origin in the hyperbolic space. Conversely, a specific concept lies further.

To push a specific concept $q$ within the aperture $\omega(q)$, a penalty is added by the angular residual of outward point $p$ with an exterior angle $\phi(p, q)$, formulated as:

$$\mathcal{L}_{ent}(p, q) = \max\big(0, \ \phi(p, q) - \eta\omega(q)\big), \tag{6}$$

where $\eta$ is a threshold making $\omega(q)$ flexible to fit $p$ at different spatial distances from $q$. The exterior angle $\phi(p, q)$ is written as:

$$\phi(p, q) = \cos^{-1}\Big(\frac{p_0 + q_0\kappa\langle p, q\rangle_{\mathcal{L}}}{\|\tilde{q}\|\sqrt{(\kappa\langle p, q\rangle_{\mathcal{L}})^2 - 1}}\Big). \tag{7}$$

The entailment cones aim to consider both intra-modality entailments and inter-modality entailments, we formulate the CAD-aware hierarchical entailment $\mathcal{L}_{\text{Che}}$ as:

$$\mathcal{L}_{\text{Che}}(I, T, I_{box}, T_{box}) = \frac{1}{4}\Big(\mathcal{L}_{ent}(I^{box}, T^{box}) + \mathcal{L}_{ent}(I, T) + \mathcal{L}_{ent}(I, I^{box}) + \mathcal{L}_{ent}(T, T^{box})\Big). \tag{8}$$

## 4 EXPERIMENTS

### 4.1 EXPERIMENTAL SETUP

**Datasets**. We use the Text2CAD (Khan et al., 2024b) dataset, which contains approximately 680k text prompts, ranging from abstract to expert levels. Basic prompts and expert prompts are used for our training of prompt enhancement. Then we obtain isometric images from the DeepCAD (Wu et al., 2021) dataset and convert them to sketch-style without any color. Besides, for an image-text pair, we first use spaCy (Honnibal et al., 2020) to extract non-abstract noun phrases from the text into a list. For the image, we use the GLIP model (Li et al., 2022) to predict the bounding boxes of the entities in the CAD image. This results in approximately 150k training samples and about 16k test and validation samples. Each sample contains multiple pairs of image boxes and text boxes.

**Training Details**. Our transformer consists of 4 encoder blocks and 4 decoder blocks with 8 attention heads. The learning rate is 0.001 with the Adam optimizer. Dropout is 0.1. The maximum number of word tokens $N_T$ is fixed as 512, and CAD tokens $N_c$ as 60, the images are resized to a size of $224 \times 224$. The CAD sequence embedding dimension $d$ is 256. The transformer has been trained for 100 epochs using 1 Nvidia 4090 GPU for 4 days. Then we train a decoder-only structure which takes the random generation latent vector $z$ and the image-test pairs as input for 100 epochs.

**Evaluation Metrics**. To examine the performance of our method, we measure the quality of both CAD sequences and the generated 3D CAD geometry. For CAD sequences, we use the command accuracy ($Acc_{cmd}$) and the parameter accuracy ($Acc_{para}$) of all command types, including line, arc, circle, extrusion, and the average parameter accuracy. For 3D geometry, we use Chamfer Distance (CD) to measure the difference of 3D CAD models between our results and ground truth, and Invalid Ratio (IR) for the model's unavailability rate.

### 4.2 EVALUATION

Our evaluation spans four aspects. First, we compare our method with five text- or image-conditioned generators: Img2CAD (Chen et al., 2025b), Text2CAD, DeepCAD + T, DeepCAD + I, and DeepCAD + T&I to assess gains. Second, we perform a modality ablation, disabling the text and image in turn to measure the contributions. Third, we test the prompt-enhancement module by removing it or substituting a generic LLM expansion, evaluating how wording precision influences quality. Fourth, we ablate the two CAD-aware learning $\mathcal{L}_{\text{Chc}}$ and $\mathcal{L}_{\text{Che}}$ separately and together, then substitute them with $\mathcal{L}_{\text{InfoNCE}}$, observing the effect on structural coherence of the CAD models.

**Comparison with other effective methods**.

We compare our multi-modal generator with five competitive baselines trained under identical settings: Img2CAD, Text2CAD, DeepCAD with textual conditioning (DeepCAD + T), DeepCAD with image conditioning (DeepCAD + I), and DeepCAD with both textual and image conditioning (DeepCAD + T&I). Since the code of Img2CAD is not publicly available, we use the performance metrics reported directly in their paper. The Text2CAD setting uses basic- and expert-level descriptions.

Table 1: Performance comparisons of our model with other methods.

| Method | Acc_para↑ | | | | | Acc_cmd↑ | CD↓ | | IR↓ | Condition |
|---|---|---|---|---|---|---|---|---|---|---|
| | Line | Arc | Circle | Extrusion | Avg | | Median | Mean | | |

I : Image    B : Basic text    E : Expert text

| Method | Line | Arc | Circle | Extrusion | Avg | $Acc_{cmd}$↑ | Median | Mean | IR↓ | Condition |
|---|---|---|---|---|---|---|---|---|---|---|
| Img2CAD | - | - | - | - | 68.77 | 80.57 | **0.16** | - | 28.8 | I |
| DeepCAD | 78.01 | 6.27 | 67.46 | 90.38 | 60.53 | 77.03 | 77.59 | 152.60 | 10.13 | I |
| DeepCAD | 80.13 | 19.42 | 75.57 | 82.33 | 64.36 | 80.09 | 31.47 | 102.81 | 7.35 | E |
| Text2CAD | 79.25 | 8.01 | 71.04 | 93.66 | 63.00 | 79.15 | 74.20 | 150.15 | 9.91 | B |
| Text2CAD | 85.40 | **41.52** | 80.18 | 96.24 | 75.84 | 82.82 | 0.45 | 29.29 | 2.41 | E |
| DeepCAD | 85.55 | 40.24 | 82.16 | **97.47** | 76.36 | **85.67** | 0.43 | **21.72** | 2.18 | I E |
| **Ours** | **86.74** | 41.22 | **84.48** | 95.73 | **77.04** | 84.47 | 0.23 | 23.82 | **1.57** | I B |

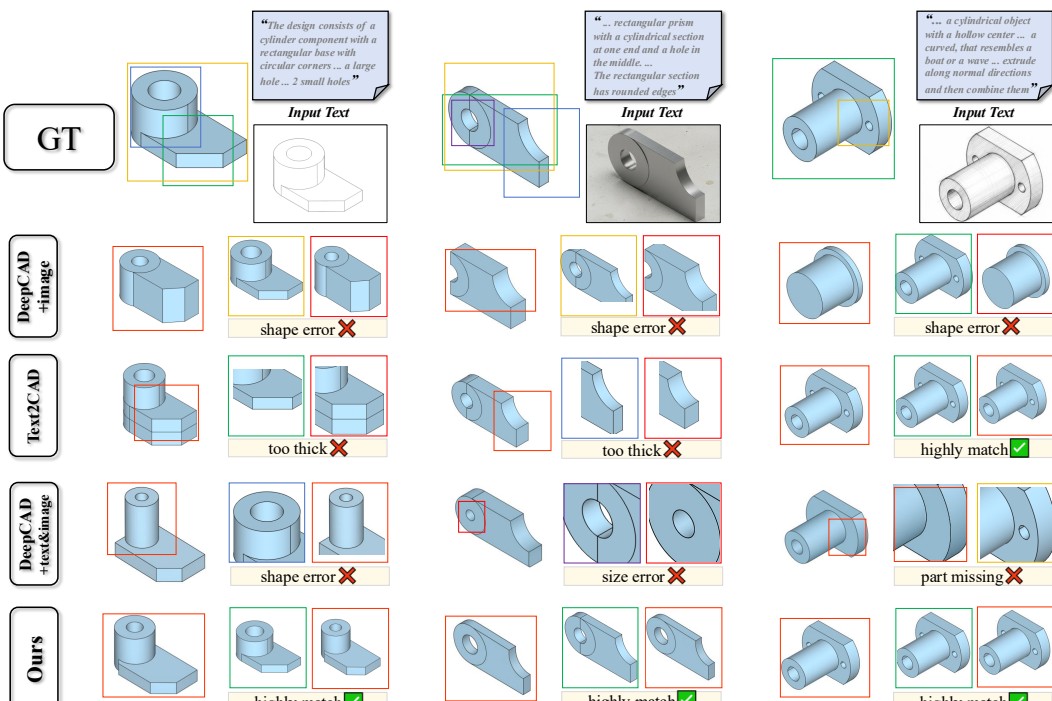

Figure 4: Our comparison with other effective methods. The red boxes represent the boxes of the comparison methods and our method, while the boxes in other colors correspond to the ground truth (GT). Sample 1 is an isometric CAD view processed to keep only outlines. Samples 2 and 3 are CAD images generated by Gemini 2.5 Flash Image in real-world and hand-drawn styles, respectively. Our method demonstrates excellent results across multiple styles of image inputs.

DeepCAD + T extends the original DeepCAD framework by incorporating expert-level language descriptions as additional input, allowing the model to generate designs based on text prompts. DeepCAD + I augments DeepCAD with images, allowing it to perform conditional CAD generation on visual outlines. DeepCAD + T&I incorporates both expert-level text prompts and images. Our model takes basic-level text prompts and single-view images as input, which are much more accessible than lengthy text or multiview images in a single modality.

Figure 4 presents the qualitative comparison. Assemblies produced by our method display well-resolved edges, correct counts of fine features, and proper alignment between mating parts. In contrast, DeepCAD + I sometimes omits small features that are faint in the image. Text2CAD with expert-level prompts occasionally introduces primitives not implied by the prompt, reflecting a weaker alignment between linguistic semantics and geometry. DeepCAD + T&I occasionally misjudges sizes and omits details, pointing to difficulties in aligning texts and images well. These visual differences highlight the advantage of anchoring the language to an explicit geometric prior.

Table 2: Ablation studies on each component of our proposed method.

| Method | $Acc_{para}\uparrow$ | | | | | $Acc_{cmd}\uparrow$ | CD$\downarrow$ | | IR$\downarrow$ |
|---|---|---|---|---|---|---|---|---|---|
| | Line | Arc | Circle | Extrusion | Avg | | Median | Mean | |
| Ablation Study 1: Modality Usage  **1** Text  **2** Image | | | | | | | | | |
| Basic model w/ **1** | 80.70 | 32.25 | 76.68 | 89.13 | 69.69 | 77.55 | 1.87 | 35.06 | 5.38 |
| Basic model w/ **2** | 81.23 | 31.76 | 78.09 | 88.14 | 69.81 | 78.12 | 1.94 | 37.42 | 5.42 |
| Ablation Study 2: CPEM  **1** Basic text  **2** Vanilla LLM  **3** SFT LLM | | | | | | | | | |
| Basic model w/ **1** | 83.28 | 36.03 | 81.75 | 93.64 | 73.68 | 80.04 | 0.45 | 25.51 | 2.39 |
| Basic model w/ **2** | 84.10 | 38.78 | 83.46 | 94.29 | 75.16 | 82.38 | 0.35 | 25.04 | 2.24 |
| Ablation Study 3: CAD-Aware Learning  **1** $\mathcal{L}_{\text{InfoNCE}}$  **2** $\mathcal{L}_{\text{Chc}}$  **3** $\mathcal{L}_{\text{Che}}$ | | | | | | | | | |
| Basic model | 83.02 | 36.69 | 81.43 | 91.71 | 73.21 | 80.60 | 0.54 | 30.85 | 2.62 |
| Basic model w/ **1** | 84.52 | 39.19 | 82.93 | 93.21 | 74.96 | 82.18 | 0.46 | 27.01 | 2.24 |
| Basic model w/ **2** | 84.08 | 38.35 | 82.72 | 93.41 | 74.64 | 81.91 | 0.47 | 26.05 | 2.27 |
| Basic model w/ **3** | 84.37 | 38.29 | 82.50 | 94.61 | 74.94 | 82.29 | 0.38 | 25.26 | 2.19 |
| **Ours( 1 2 3 2 3 )** | **86.74** | **41.22** | **84.48** | **95.73** | **77.04** | **84.47** | **0.23** | **23.82** | **1.57** |

Table 1 confirms the visual trends. Compared with Text2CAD w/ basic text prompt, our approach improves the average parameter accuracy by more than 10% and the command accuracy by 6% and reduces the Chamfer distance significantly. Compared to Text2CAD with expert-level text input, we attain extremely close or even better results using much shorter text as input, indicating a tighter adherence to both semantic intent and geometric ground truth. The gains illustrate that the integration of text and image cues, together with CAD-aware objectives, is essential to produce models that are precise, complete, and ready for downstream manufacturing.

**Practicality Analysis.** We conduct a practical evaluation of our method in Figure 8 of the Appendix, demonstrating that it achieves better CAD model reconstruction results on both real-world photographs and hand-drawn images.

**Ablation Study of Modality Usage**.

We isolate each input modality to see how it affects the quality of the generation. Three variants are tested: a text-only model conditioned on the expanded prompt, a contour-only model guided solely by the rough sketch, and our full multimodal system that fuses both in the latent space.

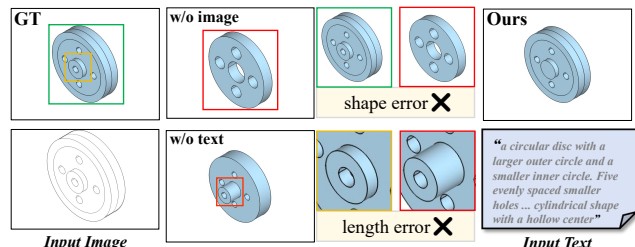

Figure 5: Ablation Study of Modality Usage.

Figure 5 highlights the visual outcome. The multimodal version retains crisp edges, complete feature sets, and correct part relationships, indicating effective integration of linguistic and visual cues. In contrast, the text-only runs sometimes misinterpret relative scales-e.g., missized holes or fillets-revealing limitations in extracting spatial context from language alone. In the condition, contour-only, relying solely on sketch input, occasionally omits subtle features that lack clear visual prominence, suggesting difficulty in resolving ambiguous or underspecified geometry. These results emphasize the importance of combining modalities to capture both intent and detail.

The Ablation Study 1 part in Table 2 back up the visuals. Compared to single-modality generation based on text or images, joint guidance raises the average parameter accuracy by more than 11% and the command accuracy by 8% over our method using single-modality, confirming that the fused model generates sequences that are both syntactically valid and geometrically exact.

**Ablation Study of CPEM**. To gauge how the wording of the prompt guides the model, we tested three settings. First, we directly use an basic-level text prompt for generation. Second, we employ a Generic-LLM to expand the prompt, providing additional descriptions. Third, we fine-tune a large model with expert-level CAD descriptions, allowing it to better understand CAD terminology and generate more accurate expansions aligned with design intent.

Figure 6 makes the differences clear. Basic-level text prompts leave room for interpretation, so the decoder often guesses sizes or depths and occasionally places parts in the wrong order. Generic-LLM expansion adds details, but such details are sometimes of the wrong kind: informal wording or loosely defined dimensions push the decoder toward ambiguous commands, leading to misplaced features or unnecessary primitives. The expert-refined prompt conversely uses vocabulary that the decoder associates with concrete operations and terms. These phrases tie directly to the latent grammar learned by the auto-encoder, so the decoder draws the feature with an accurate size.

The Ablation Study 2 part in Table 2 confirm the visual impression. Expert-SFT improves the parameter accuracy on average by more than 3% over the generic expansion and by more than 5% compared to the basic-level text, and the median CD drops by nearly half. These gains highlight a simple lesson: without domain-aware wording, extra text can be more noise than help, but with the

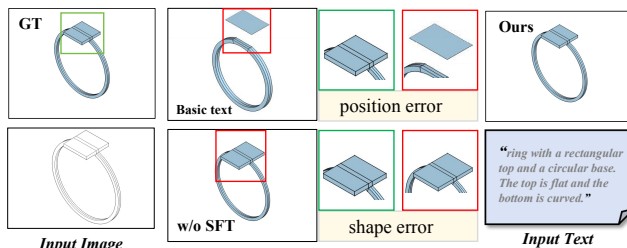

Figure 6: Ablation Study of CPEM.

right technical language, the prompt becomes a reliable guide, steering the latent code toward assemblies that satisfy both geometric rules and user intent.

**Ablation Study of CAD-Aware Learning**.

To clarify the impact of our CAD-aware Learning, we run four variants: (i) without both losses; (ii) with the Information Noise Contrastive Estimation Loss $\mathcal{L}_{\text{InfoNCE}}$ (Described in Appendix A.6); (iii) without the Hierarchical Contrastive Learning loss $\mathcal{L}_{\text{Chc}}$; (iv) without the Hierarchical Entailment Learning loss $\mathcal{L}_{\text{Che}}$. $\mathcal{L}_{\text{Chc}}$ encourages a coarse-to-fine correspondence between text, image features, and latent CAD primitives, while $\mathcal{L}_{\text{Che}}$ teaches the model that higher-level assemblies must logically subsume child components across modalities.

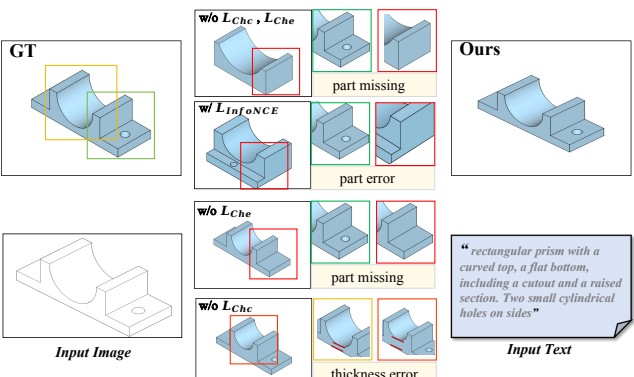

Figure 7: Ablation Study of CAD-Aware Learning.

Figure 7 illustrates the qualitative impact of each loss component. When the cross-hierarchy consistency loss $\mathcal{L}_{\text{Chc}}$ is removed, the alignment between modalities deteriorates, often resulting in sibling parts being misplaced. In contrast, removing the hierarchical enforcement loss $\mathcal{L}_{\text{Che}}$ preserves local arrangements but disrupts the overall structural coherence. Omitting both losses or substituting them with $\mathcal{L}_{\text{InfoNCE}}$ amplifies these issues, producing outputs with disjointed or non-manifold geometry.

The Ablation Study 3 part in Table 2 provide a quantitative view. Both omitting $\mathcal{L}_{\text{Chc}}$ and removing $\mathcal{L}_{\text{Che}}$ cause an average fall of 3% in average parameter accuracy, reflecting misbound primitives and structural incoherence. The complete model retains the best scores on every metric, confirming that contrastive alignment and entailment regularization work together: they align modalities at multiple scales and guide the latent space toward structurally faithful CAD generations.

## 5 CONCLUSION

In this paper, we introduce HMFusion, a multimodal paradigm for generating CAD models that reduces the reliance on detailed text and increases the accuracy of the CAD models. Our method introduces a multistage generation strategy and a bi-modal alignment and fusion mechanism to enhance the model's ability to work with brief inputs and to improve the semantic-geometric consistency of the generated CAD models. Through comprehensive evaluations, we demonstrate significant improvements over existing methods.

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

# A APPENDIX

## A.1 ETHICS STATEMENT

This work adheres to the ICLR Code of Ethics. In this study, no human subjects or animal experimentation was involved. All datasets used, including Text2CAD datasets, were sourced in compliance with relevant usage guidelines, ensuring no violation of privacy. We have taken care to avoid any biases or discriminatory outcomes in our research process. No personally identifiable information was used, and no experiments were conducted that could raise privacy or security concerns. We are committed to maintaining transparency and integrity throughout the research process.

## A.2 REPRODUCIBILITY STATEMENT

We have made every effort to ensure that the results presented in this paper are reproducible. All code and datasets will be made publicly available in an repository to facilitate replication and verification when the paper is accepted. The experimental setup, including training steps, model configurations, and hardware details, is described in detail in the paper.

Additionally, datasets we used, such as Text2CAD datasets, are publicly available, ensuring consistent and reproducible evaluation results.

## A.3 USE OF LLMS

Large Language Models (LLMs) were used to aid in the writing and polishing of the manuscript. Specifically, we used an LLM to assist in refining the language, improving readability, and ensuring clarity in various sections of the paper. The model helped with tasks such as sentence rephrasing, grammar checking, and enhancing the overall flow of the text.

It is important to note that the LLM was not involved in the ideation, research methodology, or experimental design. All research concepts, ideas, and analyses were developed and conducted by the authors. The contributions of the LLM were solely focused on improving the linguistic quality of the paper, with no involvement in the scientific content or data analysis.

The authors take full responsibility for the content of the manuscript, including any text generated or polished by the LLM. We have ensured that the LLM-generated text adheres to ethical guidelines and does not contribute to plagiarism or scientific misconduct.

## A.4 COMMAND PARAMETER REPRESENTATION

The full parameter vector for each command is $p_i = [x, y, \alpha, f, r, \theta, \phi, \gamma, p_x, p_y, p_z, s, e_1, e_2, b, u]$. As detailed in Sec.3, these parameters are normalized and quantized.

Initially, all CAD models are resized to fit within a bounding box of dimensions $2 \times 2 \times 2$ (without translation) such that all attributes are well-bounded. Specifically, the sketch plane's origin $(p_x, p_y, p_z)$ and the bidirectional extrusion lengths $(e_1, e_2)$ are constrained within the interval $[-1, 1]$. Profile scale values are limited to the range $[0, 2]$, and the orientation angles $(\theta, \phi, \gamma)$ are confined to $[-\pi, \pi]$.

Next, each 2D sketch is rescaled to lie within a unit square, such that its reference point—typically the bottom-left vertex—maps to the center $(0.5, 0.5)$ of the square. As a result, endpoint coordinates $(x, y)$ and circle radius $r$ fall into the $[0, 1]$ range. The sweep angle of any arc is similarly bounded within $[0, 2\pi]$.

Then, continuous variables are quantized into 256 levels and encoded as 8-bit integers. Discrete parameters are retained in their original form. For instance, the arc direction flag $f$ uses 0 to indicate clockwise and 1 for counter-clockwise motion. The constructive solid geometry (CSG) operation type $b \in \{0, 1, 2, 3\}$ encodes *new body, union, subtraction*, and *intersection*, respectively. Meanwhile, the extrusion mode $u \in \{0, 1, 2\}$ corresponds to *one-sided, symmetric*, and *two-sided* extrusion types.

The command accuracy $Acc_{cmd}$ and parameter accuracy $Acc_{para}$ are defined as follows:

$$\text{ACC}_{\text{cmd}} = \frac{1}{N_c} \sum_{i=1}^{N_c} \mathbb{I}[t_i = \hat{t}_i], \qquad \text{ACC}_{\text{param}} = \frac{1}{K} \sum_{i=1}^{N_c} \sum_{j=1}^{|\hat{p}_i|} \mathbb{I}[|p_{i,j} - \hat{p}_{i,j}| < \eta] \mathbb{I}[t_i = \hat{t}_i]. \qquad (9)$$

The proposed multimodal CAD generator lowers the barrier to precise modelling by letting engineers and casual users produce manufacturable designs from a brief prompt and a sketch, accelerating concept iteration and reducing reliance on expert drafting skills in industrial workflows. At the same time, it lays the groundwork for future studies on automatic parametric refinement, assembly-level constraint propagation, and seamless integration with mainstream CAD toolchains, paving the way for even more efficient and adaptive design processes.

### A.5 NETWORK ARCHITECTURE AND TRAINING DETAILS

**Autoencoder.**Our Transformer-based encoder and decoder consist of four sequential layers. Each layer incorporates eight attention heads and a feed-forward network with a hidden size of 512. We apply layer normalization and introduce a dropout rate of 0.1 within every Transformer block.

Following the final block in the decoder, two distinct linear projection layers are applied. One predicts the type of CAD command with weights $W_1 \in \mathbb{R}^{256 \times 6}$, while the other generates the associated command parameters with weights $W_2 \in \mathbb{R}^{256 \times 4096}$. The resulting 4096-dimensional vector is then reshaped into a matrix of shape $16 \times 256$, representing each of the 16 parameter slots.

**Latent-GAN.**As described in Section 3.2, the latent-CAD is employed to produce latent vectors for CAD command generation. Both the generator and discriminator architectures follow MLP networks, consisting of four hidden layers with 512 neurons each. The generator receives a 64-dimensional random noise vector as input and outputs a latent vector of 256 dimensions.

For training, we employ the WGAN-gp approach. During optimization, the discriminator (critic) is updated five times per generator iteration. A gradient penalty term with a weight of 10 is added to enforce smoothness. Training is performed over 200,000 steps with a mini-batch size of 256. Adam optimizer is used with a learning rate of $2 \times 10^{-4}$ and $\beta = 0.5$.

### A.6 IMPLEMENTATION DETAILS

As evaluated by (Atigh et al., 2022), setting curvature $\kappa$ as a learnable parameter yields the best performance, so we follow them and keep the curvature a learnable parameter with an initial value of $\kappa = 1.0$ and clamped within $[0.1, 10.0]$. In the prompt enhancement section, we set LoRA rank to be 8 and $\alpha$ to be 32 during fine-tuning.

In the ablation study section, the InfoNCE loss is defined as:

$$\mathcal{L}(I, T) = -\frac{1}{N} \sum_{i=1}^{N} \log \left( \frac{\exp\left(\frac{I_i \cdot T_i}{\tau}\right)}{\sum_{j=1}^{N} \exp\left(\frac{I_i \cdot T_j}{\tau}\right)} \right), \qquad (10)$$

$$\mathcal{L}_{\text{InfoNCE}} = \frac{1}{2}\big(\mathcal{L}(I, T) + \mathcal{L}(T, I)\big). \qquad (11)$$

When we compare with other methods, we employ the same pre-trained visual encoder, ViT-L/14-336 (Dosovitskiy et al., 2020), and map its output to the same latent space for DeepCAD + T&I and DeepCAD + I. For Text2CAD and DeepCAD + T&I, we employ the same pretrained textual encoder, BERT encoder (Devlin et al., 2019).

### A.7 OUR GENERATED CAD MODELS

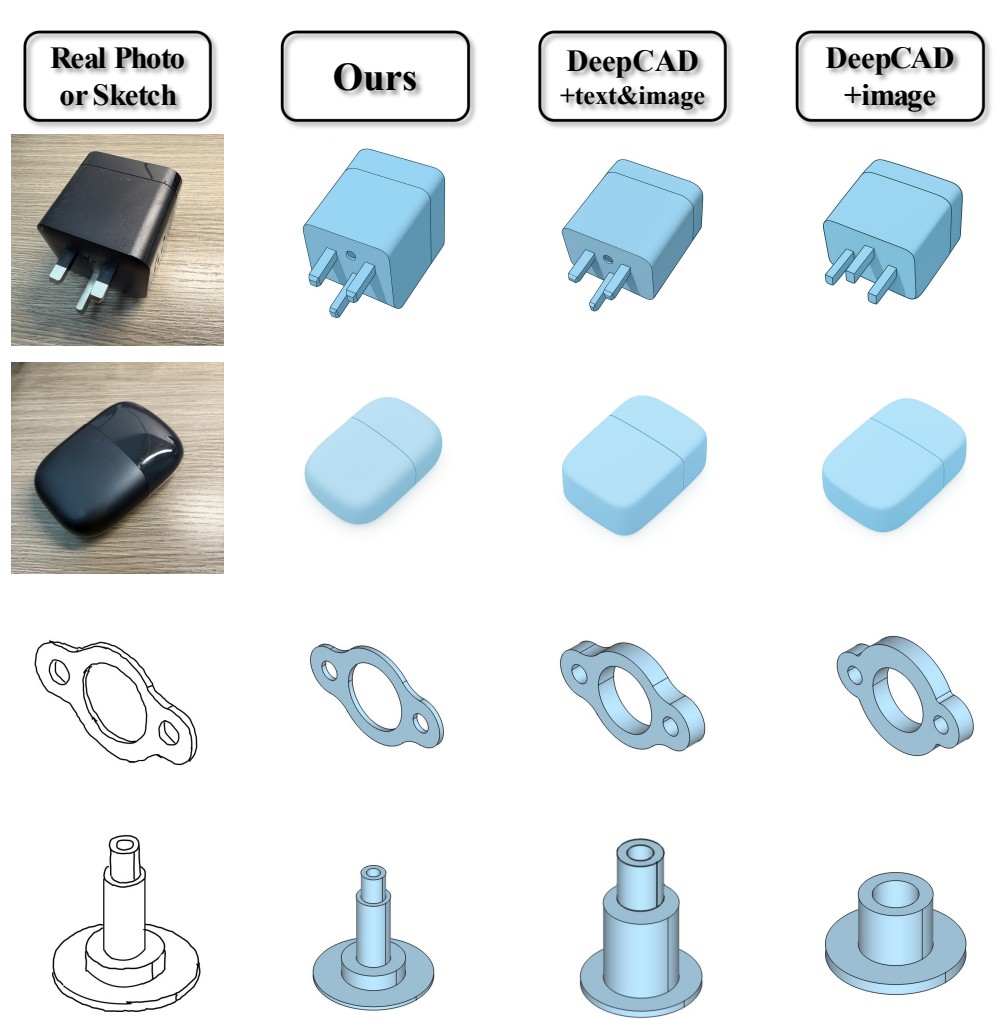

Figure 8: Visual comparison with real-world photos and hand-drawn sketches as input. Our method generates excellently matched CAD models, demonstrating great practicality in our real world.

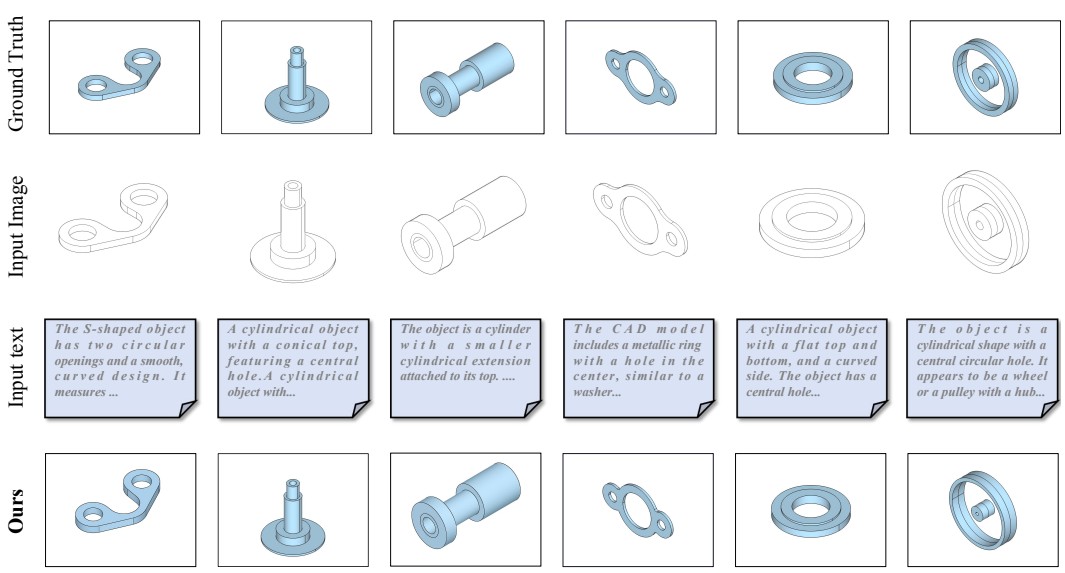

Figure 9: Some cases of our generated CAD models.

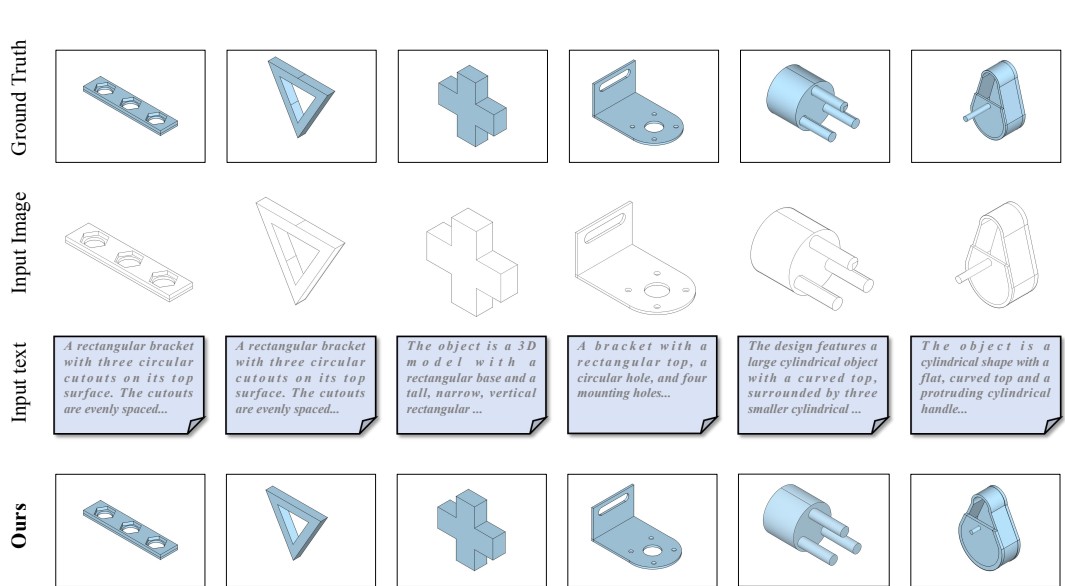

Figure 10: Some cases of our generated CAD models.

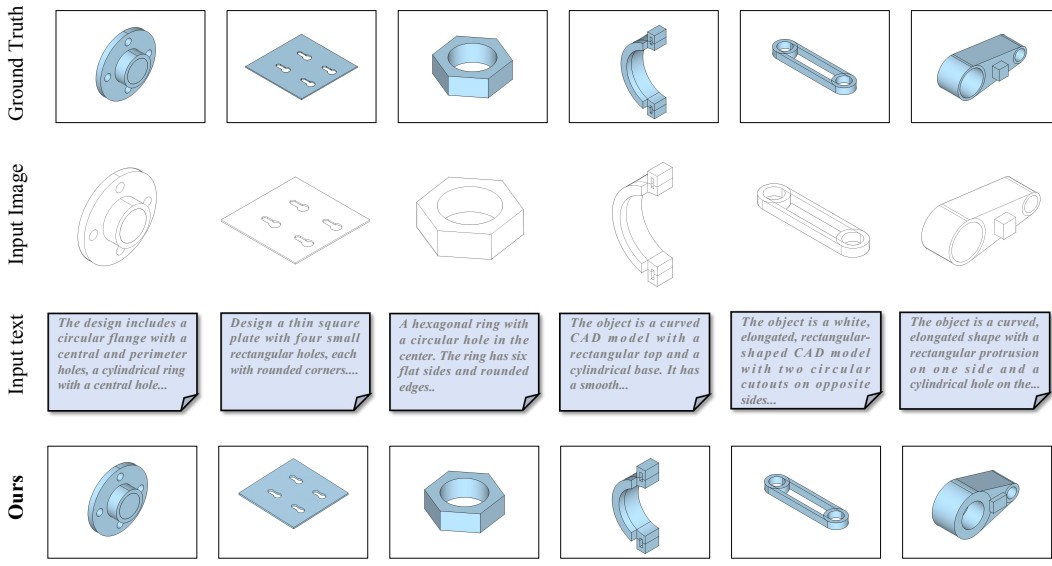

Figure 11: Some cases of our generated CAD models.

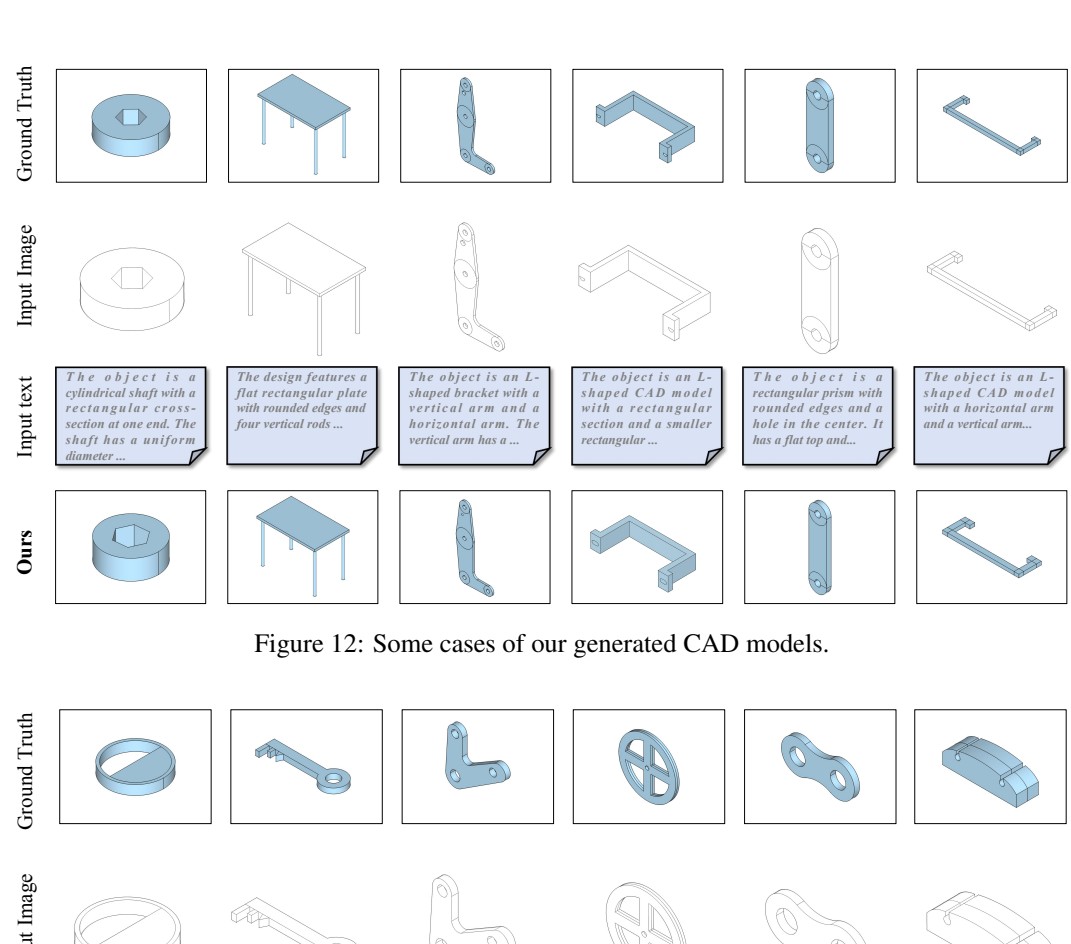

Figure 12: Some cases of our generated CAD models.

Figure 13: Some cases of our generated CAD models.

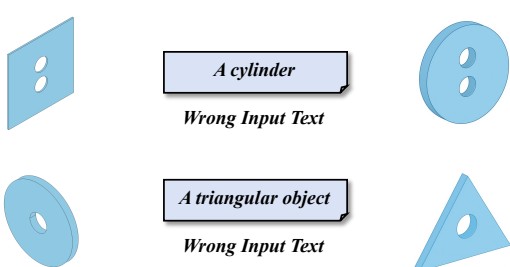

Figure 14: Generation results under modality conflict. When visual and textual cues contradict each other, the model cannot favor either modality, producing unstable and semantically inconsistent outputs.

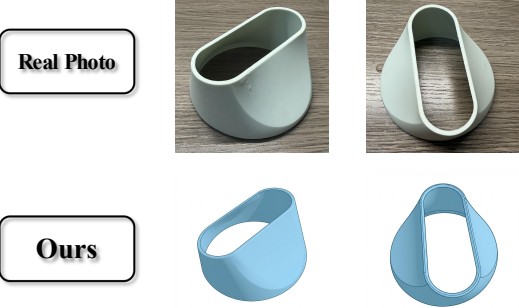

Figure 15: Qualitative results of CAD model generation from real-world images captured under different viewpoints. Our method produces consistent and structurally accurate CAD reconstructions despite significant view-angle variations.

