# OpenReview forum: "HMFusion: Hierarchical Multi-Modality Fusion for CAD Representation"
_ICLR.cc/2026/Conference — ICLR 2026 Conference Withdrawn Submission_

### Official Review · Reviewer_K9p3 · 2025-10-23

**Soundness:** 2
**Presentation:** 2
**Contribution:** 2
**Rating:** 4
**Confidence:** 4

**Summary:**

This paper proposes a novel multimodal framework for generating accurate CAD models from minimal inputs—a brief text prompt and a single sketch or image. It enhances user prompts into detailed CAD descriptions using a fine-tuned LLM. Also it proposes a hierarchical multi-modality fusion module to fuse textual and visual features into a hyperbolic space, in order to ensure geometric consistency and semantic fidelity.

**Strengths:**

1. Main idea presentation of this paper and pipeline figure are easy to understand.

2. The proposal of learning relationship between primitives and entire CAD models sounds interesting in Line 259-260.

**Weaknesses:**

Major:

1. This paper may not be well completed in terms of writing, as Figure 3—which illustrates the core pipeline—is never referenced in the main text. Moreover, the "Modal Enhancement" module shown in the top-right of Figure 3 is never elaborated upon in the main body or even in the appendix.

2. While the idea sounds interesting, the claim in lines 259–260 regarding learning the relationship between primitives and entire CAD models does not seem fully justified. This relationship is only learned in the “image + text” fusion part, and doesn't include command sequence. Primitives such as loop, profile, and solid can exist in the command sequence, according to HNC-CAD:

- Hierarchical Neural Coding for Controllable CAD Model Generation, Xu et al. (https://arxiv.org/abs/2307.00149)

This explicit primitive information from command sequence is not used.  The fusion module claims to learn relationship between primitives and entire CAD models, but only relies on “image + text”. The reviewer is concerned about the correctness of this claim. How to prove the correctness of this claim if so?

3. The motivation to utilize hyperbolic space technique for multi-modal fusion in this paper is not clearly elaborated in Abstract and Introduction.

4. In Table.1, the evaluation result of "Ours", with "image + expert text" as input, seems missing. The reviewer understands the paper trys to propose use minimal text prompt. The reviewer still thinks it's essential to present the result of  "image + expert text".

5. In Table.1, the multi-modal input of DeepCAD and "Ours" do not align, which may introduce improper comparison.

6. In Table 2, the “Modality Enhancement” ablation appears inconsistent with the rest of the paper: the “Modal Enhancement” module shown in the top-right of Figure 3 is never described in the main text or the appendix, and this component is not explicitly listed among the paper’s contributions.

Minor:

1. Math definition in L236 has problem: $g_M(M_k) = \exp_{\mathbf{0}}^{\kappa}(f_M(M_k)$

Should complete the bracket here:  $g_M(M_k) = \exp_{\mathbf{0}}^{\kappa}(f_M(M_k))$

2. The citation formatting seems a bit off. For example, in line 113:

> such as point cloudsMa et al. (2024); Wu et al. (2021); ...

There should be a space before the citation, i.e., “point clouds (Ma et al., 2024; Wu et al., 2021)”. The same issue appears throughout the paper.

3. Some recent works on CAD generation are missing from the related work section. This area is evolving rapidly, particularly with the rise of LLM-based methods, so including a more comprehensive coverage would strengthen the paper’s survey:

- CAD-MLLM: Unifying Multimodality-Conditioned CAD Generation With MLLM, Xu et al. (https://arxiv.org/abs/2411.04954)

- CAD-Llama: Leveraging Large Language Models for Computer-Aided Design Parametric 3D Model Generation, Li et al. (https://arxiv.org/abs/2505.04481)

- CAD-Coder: Text-to-CAD Generation with Chain-of-Thought and Geometric Reward, Guan et al. (https://arxiv.org/abs/2505.19713)

- CADmium: Fine-Tuning Code Language Models for Text-Driven Sequential CAD Design, Govindarajan et al. (https://arxiv.org/abs/2507.09792)

**Questions:**

1. What is the real effectiveness of the hyperbolic space technique in multi-modal fusion? This is the reviewer's core concern about this paper's contribution. It is recommended to include a comparison against standard multi-modal fusion methods to showcase the essentiality of this module and enhance the paper's technical soundness. And also to highlight the motivation of utilizing hyperbolic space for multi-modal fusion.

---

> ### Author Response · Authors · 2025-11-21
> **Response to Weaknesses and Questions**
>
> We sincerely appreciate your valuable feedback. There may exist somewhere you misunderstood or we gave a vague description in the paper. We will explain these points you mentioned in detail. The explanations and modifications in the rebuttal have been marked in blue in our revised paper.
>
> ### W1&W6. Explanation about “Modality Enhancement”
>
> Actually, we mentioned “The Modality Enhancement” in the **first paragraph in 3.2 CAD GENERATION FROM MULTI-MODALITY CONDITIONS**. The Modality Enhancement refers to the SFT-LLM in this paper. This module is relatively simpler, hence it has not been elaborated in detail.
>
> ### W2. We do not learn the relationship between primitives and entire CAD models.
>
> On the one hand, it will bring more computational cost if we include the learning of the relationship between primitives and entire CAD sequences. On the other hand, currently there exists no appropriate method that segments the CAD sequences to sub-primitives in a suitable way like spaCy and GLIP do to text and images.
>
> ### W3. The motivation for the utilization of hyperbolic space.
>
> The motivation for using hyperbolic space is its natural suitability for modeling **hierarchical structures**, which are inherent in CAD features and their multi-modal representations. CAD models often form tree-like part hierarchies, and unlike Euclidean embeddings that tend to distort such hierarchies, hyperbolic space preserves these relationships with **minimal distortion**. When we describe a CAD model with texts or images, they all consist of primitives, such as a word in the text, a part of the model in the image. The hyperbolic space allows us to embed both textual and visual features in a unified manifold that mirrors their hierarchical organization. These properties explain why hyperbolic geometry is especially effective for capturing CAD feature hierarchies and aligning cross-modal structures, resulting in a more consistent and accurate fusion than standard approaches.
>
> ### W4. Using expert prompts is not necessary.
>
> We did not feed the expert-level prompts to our model, since such a detailed and lengthy description for a CAD model is unrealistic and almost impossible to be told accurately by a human. Our model not only aims to generate accurate CAD sequences, but also provides **a more accessible way for CAD beginners** to generate with relatively brief prompts. However, we feed expert prompts to other models, and experiment shows our model outperforms them under such situations.
>
> ### W5. The equal comparison with DeepCAD.
>
> The input modalities of DeepCAD are the same as ours, both with images and texts, as shown in the 6th line in the **Table 1**.
>
> ### Q1. The real effectiveness of hyperbolic space.
>
> The hyperbolic space technique has proven highly effective in our multi-modal fusion framework, as evidenced by both ablation studies and overall performance gains. In particular, when we remove the hyperbolic hierarchical alignment or replace it with InfoNCE loss, we observe a clear drop in performance **(Table 2)**, along with misbound CAD primitives, compared to the full model. This indicates that the hyperbolic embedding contributes significantly to capturing the underlying cross-modal relationships.

---

> > ### Comment · Reviewer_K9p3 · 2025-11-25
> >
> > The reviewer appreciates the author's feedback.
> >
> > ## W1&W6. Explanation about “Modality Enhancement”
> >
> > Thanks for your elaboration to add more reference in 3.2. But Figure 3 is still not referred to in main body, and the link between 3.2 and Modality Enhancement in Figure 3 is not obvious. Explicitly pointing out the relationship will make the paper easier to follow.
> >
> > ## W2. We do not learn the relationship between primitives and entire CAD models.
> >
> > If so, I can't support the statement of "Considering the local-global relationship in image and text, we use hierarchical
> > Compositional Entailment to learn the relationship between primitives and entire CAD models." listed in current L258-260.
> >
> > ## W3. The motivation for the utilization of hyperbolic space.
> >
> > Could you please help to highlight why "The hyperbolic space allows us to embed both textual and visual features in a unified manifold that mirrors their hierarchical organization." Or is that enough to explain it, just with better results in ablation from table2?
> >
> > ## W4. We did not feed the expert-level prompts to our model, since such a detailed and lengthy description for a CAD model is unrealistic and almost impossible to be told accurately by a human.
> >
> > Do you mean that expert-prompted text-to-cad task is not necessary, like the task in Text-To-CAD and lots of other following works. If so, it might not necessary for the authors to place the result of this task as previously mentioned by authors:
> >
> > > However, we feed expert prompts to other models, and experiment shows our model outperforms them under such situations.
> >
> > ## W5. The equal comparison with DeepCAD.
> >
> > The input of DeepCAD is expert text. I still can't agree that this comparison is fair. (Different texts will lead to different difficulties)
> >
> > ## Q1. The real effectiveness of hyperbolic space.
> >
> > The ablation comparison is good to me. But I'm not completely convinced by "capturing the underlying cross-modal relationships".

---

> ### Author Response · Authors · 2025-11-28
> **Further response 1/2**
>
> Reviewer K9p3:
>
> Thank you again for your detailed reply. We have made further response based on your questions. If you have any further questions, please feel free to ask. We hope this helps you better understand our contributions and fosters a positive attitude towards our article.
>
> ### W1&W6. Clarification of the “Modality Enhancement” Module.
>
> Thank you for pointing this out. In the latest updated paper, Figure 3 has been explicitly mentioned. We have renamed "Modality Enhancement" to "CAD Prompt Enhancement Module (CPEM)" and listed it separately in Section 3.2 for easier following.
>
> ### W2. Scope of Hierarchical Representation Learning.
>
> We thank the reviewer for the clarification. However in our paper, the term **primitive** refers to the **sub-components of the image or text**, not to explicit CAD parts or geometric entities. Our *Hierarchical Representation(HR)* is **not** designed to model the whole command order structure under CAD model. It intends by its use to get a better and more consistent embedding through the *input modalities* (image and text). The CAD command sequence is the **final prediction target** (and not part of the input to be encoded by hierarchical sequence). Thus, the relationship between primitives and full CAD—loops, profiles and solids in HNC-CAD—is **not what our HR module seeks to learn** nor is it essential for our formulation. Explicit primitive-level decomposition for command sequences would certainly prove a useful direction, but it lies outside the scope of our HR design. Our method also directly **creates CAD from image and/or text** rather than from pre-defined primitives. Investigating hierarchical structure *inside* CAD command sequences would be a promising addition to our arsenal, but does not affect the correctness of our HR module, which functions in order to increase cross-modal comprehension on the input side.
>
> ### W3. The motivation for the utilization of hyperbolic space.
>
> Our reasoning for hyperbolic space is to gain a **more faithful hierarchical representation** of the input modalities. CAD descriptions — be they written in text or image form — usually share multi-level structures, such as coarse global shapes → subparts → fine details. Euclidean embeddings, for example, generally flatten or distort tree-like structures; at the same time, hyperbolic geometry is known to preserve hierarchical relations with significantly lower distortion. Hyperbolic space is not used to encode CAD command hierarchies in our scenario, but rather an advantage that gives **better geometric space to represent and align the input modalities themselves**. This helps to produce more structured embeddings and the cross-modal alignment can be also enhanced. Experiments confirmed that transitioning from Euclidean space to hyperbolic space uniformly improves fusion quality in our work, thus validating its appropriateness for hierarchical modeling.
>
> ### W4. Expert prompts as input.
>
> We apologize for the misleading words. Our intention was to emphasize that expert-level textual descriptions are rarely available in real design scenarios, especially when creating new CAD models. In practice, designers typically provide a sketch and, at most, a short informal note, rather than the highly detailed expert-written text. Therefore, our main experiments focus on basic, non-expert text, which we believe better reflects practical usage. Even so, we also evaluate expert-level text by directly feeding it to our model without any retraining. As shown in the table below, expert text consistently improves performance, indicating that our framework is robust to both basic and expert textual inputs. We will revise the paper to avoid confusion.
>
> | Method   | line | arc | circle | extrusion | average | $Acc_{cmd}$ | Median CD | Mean CD | IR |
> |-------------------|-------------|-------------|-------------|-------------|-------------|-------------|-------------|-------------|-------------|
> | Ours w/ SFT-LLM text         | 86.74       | 41.22        | 84.48   | 95.73   | 77.04      |84.47   | 0.23 | 23.82 | 1.57 |
> | Ours w/ expert text              | 88.16        | 41.87        | 86.41   | 98.08  | 78.63      | 86.13 | 0.19   | 22.17 | 1.21 |

---

> > ### Author Response · Authors · 2025-11-28
> > **Further response 2/2**
> >
> > ### W5. The fair comparison with DeepCAD.
> >
> > Thank you for the suggestion. We conducted a fair comparison between DeepCAD and our method in the table below. As seen, our approach outperforms DeepCAD by a substantial margin when using both expert-level text and sketches. This further demonstrates the effectiveness of our method.
> >
> > | Method   | line | arc | circle | extrusion | average | $Acc_{cmd}$ | Median CD | Mean CD | IR |
> > |-------------------|-------------|-------------|-------------|-------------|-------------|-------------|-------------|-------------|-------------|
> > | DeepCAD            | 85.55       | 40.24       | 82.16   | 97.47  | 76.36      | 85.67 | 0.43   | 21.72 | 2.18 |
> > | Ours w/ expert text              | 88.16        | 41.87        | 86.41   | 98.08  | 78.63      | 86.13 | 0.19   | 22.17 | 1.21 |
> >
> > ### Minor Problems
> >
> > We have revised the relevant parts and added all the citations. Thank you to the reviewers for their valuable comments. Please feel free to raise any further questions or requests for revision.
> >
> > ### Q1. The real effectiveness of hyperbolic space.
> >
> > **A1:** As shown in the table below, hyperbolic space consistently yields higher accuracy than its Euclidean counterpart, demonstrating the effectiveness of using hyperbolic space in our model.
> >
> > | Method   | line | arc | circle | extrusion | average | $Acc_{cmd}$ | Median CD | Mean CD | IR |
> > |-------------------|-------------|-------------|-------------|-------------|-------------|-------------|-------------|-------------|-------------|
> > | Ours w/Euclidean space | 82.95 | 36.72 | 81.38  | 91.65     | 73.18   | 80.55       | 0.56      | 30.92   | 2.58 |
> > | Ours w/Hyperbolic space | 86.74 | 41.22  | 84.48   | 95.73   | 77.04      |84.47   | 0.23 | 23.82 | 1.57 |

---

### Official Review · Reviewer_EjXr · 2025-10-29

**Soundness:** 3
**Presentation:** 3
**Contribution:** 2
**Rating:** 6
**Confidence:** 4

**Summary:**

This paper proposes HMFusion, a framework for generating CAD models from minimal inputs: a brief text prompt and a single sketch. It addresses the limitations of existing methods, which either require expert-level prompts or lack dimensional accuracy. Specifically, a LoRA-tuned LLM enhances brief prompts into CAD-specific descriptions, while a hyperbolic space fusion module aligns linguistic and geometric semantics through contrastive and entailment learning. The fused representation guides a Transformer-based CAD decoder to produce accurate and consistent 3D models. Experiments show notable improvements in geometric accuracy, command precision, and structural fidelity over prior text or image approaches.

**Strengths:**

1. The model’s ability to generate manufacturable CAD models from concise prompts and simple sketches has strong industrial  interaction implications.

2. The paper is exceptionally well-written and easy to follow. Figures 2 and 3, in particular, provide a clear and intuitive overview of the complex architecture and data flow.

3. The paper tackles a well-defined and highly practical problem in generative design. The goal of enabling high-fidelity CAD generation from minimal, non-expert inputs (a simple sketch and a brief sentence)  is a significant step forward in usability compared to existing methods.

**Weaknesses:**

1. The method relies on the DeepCAD representation, which is limited to sketch and extrusion commands. It only represents the simple CAD model that is far from real-world CAD design . The paper does not discuss how the proposed hierarchical fusion would scale to more complex CAD operations (e.g.,sweeps, fillets, chamfers)

2.It is currently unclear how the model handles non distributed shapes or non mechanical CAD categories (OOD problem, such as free-form designs).

3.Considering the current model's design, which uses an SFT-LLM to expand the basic text, this raises the question of whether the user-prompted basic text must adhere to a specific format or paradigm. Furthermore, if the format of the basic text differs from what the SFT-LLM was trained on, how would the model's performance be affected? An in-depth discussion on this point would help to strengthen the paper's contribution.

4. Some descriptive analyses do not align with the tables. For instance, in the Ablation Study of Modality Enhancement section, it mentions, “Expert-SFT improves the parameter accuracy on average by more than 3%... ”but in Table 2, Ablation Study 2, the results for ”w/ SFT LLM“ do not seem to be listed.

**Questions:**

1. Have authors considered the performance of feeding expert text directly to the model in such an experiment? This comparison can further help demonstrate the significance of basic expansion in SFT-LLM.

2. How does the model perform when given conflicting information between the sketch and the text? Which one would be considered first?

3. What is the minimum length of basic text, and has the model been tested to produce the shortest description required for valid generation?

4. What would the effect be if the text expanded by the SFT-LLM were fed into Text2CAD?

---

> ### Author Response · Authors · 2025-11-21
> **Response to Weaknesses and Questions**
>
> Thank you for your positive comments. We would like to address each of the concerns you raised below and hope our explanations meet your expectations. The explanations and modifications in the rebuttal have been marked as blue in our revised paper.
>
> ### W1. Currently focus on accurate generation based on DeepCAD representation
>
> Current CAD generation methods primarily adopt the representation of DeepCAD, but existing approaches still struggle to produce highly intricate and complex CAD models. The objective of this paper is to surpass current methods and generate models using the DeepCAD representation more accurately.
>
> ### W2. Your concerns about OOD problems.
>
> Our model performs well even on samples whose images are obtained casually. This could effectively demonstrate the performance of our model. As shown in **Figure 8**, we took both **hand-drawn** images and **real-world** photos as input, with **various view angles**. Under such a situation, our model still demonstrates excellent generation results. This robustness when using **OOD data** proves that this weakness is unnecessary to be worried about.
>
> ### W3. The format of input text prompts is free.
>
> The user-prompted basic description does not have to adhere to a specific format. It just needs to describe the shape and parameters (length...). The effect of the SFT-LLM is to convert this prompt to a detailed prompt that consists of **specific words**, which are corresponded to CAD sequence commands such as line, extrude, etc, more easier to be understood by the model.
>
> ### W4. Position of results for “w/ SFT-LLM”.
>
> The model w/ SFT-LLM is listed in the last line in the **Table 2 (Ours)**. Compared to ours, the two models in **“Ablation Study 2”** is the use of text is basic text, text processed by vanilla LLM, or text processed by SFT-LLM (ours).
>
> ### Q1. Using expert prompts is not very necessary.
>
> We did not feed the expert-level prompts to our model, since such a detailed and lengthy description for a CAD model is unrealistic and almost impossible to be told by a human. Our model not only aims to generate accurate CAD sequences, but also provides **a more accessible way for CAD beginners** to generate with relatively brief prompts. However, we feed expert prompts to other models, experiment shows our model outperforms them under such a situation.
>
> ### Q2. Performance when meeting conflict information.
>
> A conflict between the two modalities can cause the generated output to become messy and not be biased toward either modality. Our original goal was for one modality to supplement the other with additional information. However, if the supplementary information contains conflicts, the model will still try to combine both sources as much as possible, which may result in outputs that appear strange and do not meet expectations. As shown in the added **Figure 14**, when the input of the text and the image exists conflicts, the model will generate something like the fusion of characteristics mentioned by both modalities, due to the original goal that one modality gets supplementary information from the other.
>
> ### Q3. The length of input text prompts.
>
> A few simple words can generate a precise model (e.g., a cylinder) because no specific details are required. The more detailed the description, the harder it is to achieve a perfect match; however, it also brings the model closer to meeting the complex demands of the industry.
>
> ### Q4. The effect when feeding SFT-LLM to Text2CAD.
>
> When this text is fed to Text2CAD, its effect will be better than feeding basic prompts but a little weaker than feeding expert prompts.

---

> ### Comment · Reviewer_EjXr · 2025-11-26
>
> Thanks for your rebuttal. I have no further questions and have decided to maintain my positive score. Nevertheless, I still suggest that the authors could include some visualization examples/quantitative analysis regarding Question 4 in the revised manuscript. This would be helpful to clearly see the model's performance (based on the text expanded by the SFT-LLM).

---

> ### Author Response · Authors · 2025-11-27
>
> Thank you for your further reply. We have provided some additional explanations regarding your fourth question. If you have any further questions, please feel free to ask. We hope this will further enhance your understanding and appreciation of our articles.
>
> ### Q4. The effect when feeding SFT-LLM to Text2CAD.
>
> **A4:** The following is a quantitative results analysis.
>
> | Method   | line | arc | circle | extrusion | average | $Acc_{cmd}$ | Median CD | Mean CD | IR |
> |-------------------|-------------|-------------|-------------|-------------|-------------|-------------|-------------|-------------|-------------|
> | Text2CAD w/ basic text         | 79.25       | 8.01        | 71.04   | 93.66   |63.00      |79.15   | 74.20 | 150.15 | 9.91 |
> | Text2CAD w/ SFT-LLM text         | 82.53       | 39.98| 75.69   | 89.06   | 71.82      | 80.71  | 0.57 | 40.08 | 4.12 |
> | Text2CAD w/ expert text              | 85.40        | 41.52        | 80.18   | 96.24  | 75.84      | 82.82 | 0.45   | 29.29 | 2.41 |

---

### Official Review · Reviewer_aMsY · 2025-11-01

**Soundness:** 3
**Presentation:** 2
**Contribution:** 4
**Rating:** 6
**Confidence:** 3

**Summary:**

This paper presents HMFusion, a multimodal CAD generation framework that combines text and image inputs. The method integrates three components: (1) a LoRA-tuned LLM for prompt enhancement that transforms short natural descriptions into detailed CAD-specific language; (2) a visual encoder that extracts geometric priors; and (3) a hierarchical fusion module operating in hyperbolic space to align visual and textual embeddings. The approach aims to improve both semantic fidelity and geometric consistency of CAD generation, addressing limitations of unimodal (text-only or image-only) systems.

**Strengths:**

The paper tackles the cross-modal alignment between text and image inputs, a relevant and underexplored problem in CAD generation. The motivation is clearly articulated: text-to-CAD systems often lack spatial grounding, while image-to-CAD lacks parametric precision. The proposed multimodal fusion framework provides a compelling attempt to balance these complementary weaknesses.

The work demonstrates substantial engineering effort and a non-trivial combination of components—text expansion, hyperbolic-space embedding, and CAD-aware contrastive and entailment losses. The experiments are thorough, including ablations across modalities, enhancement stages, and loss components. The visual results consistently support the claim of improved geometric completeness and semantic consistency. The approach seems technically sound and well-implemented, with quantitative gains over baselines in Chamfer distance and parameter accuracy.

**Weaknesses:**

The framework indeed involves several heavy modules (CAD autoencoder, LoRA-tuned LLM, GLIP-based contour extractor, hierarchical contrastive learning). This complexity may hinder reproducibility and scalability, especially since each submodule requires specialized training.

The experimental comparisons are acceptable. However, the authors claim that their baselines represent state-of-the-art methods. In my view, this is inaccurate, as DeepCAD and Text2CAD are no longer considered SOTA. In the text-to-CAD domain, more advanced frameworks such as CAD-Llama [1] and CADFusion [2] have demonstrated superior performance. While this may not be an intentional oversight, the phrasing is misleading as it disregards recent progress in the field.

Minor problems:
1. Use `\citep` if you want to create citations in the parentheses.
2. Annotation misalignment: \mathcal{L}{pair} and L{pair} are interchanged and can be confusing.

[1] CAD-Llama: Leveraging Large Language Models for Computer-Aided Design Parametric 3D Model Generation. CVPR 2025.

[2] Text-to-CAD Generation Through Infusing Visual Feedback in Large Language Models. ICML 2025.

**Questions:**

How are the text embeddings computed after segmentation? Are SpaCy-extracted phrases encoded via the same text encoder used in fusion, or is there a separate mapping network?

Have the authors evaluated against recent multimodal baselines (e.g., CADFusion, CAD-LLaMA)? If not, could they discuss performance expectations relative to those?

Since it is a multi-modal task, recent VLMs are also competitive baselines. Can the authors test the performance of some of these models (e.g. GPT-4o or later models) on it?

---

> ### Author Response · Authors · 2025-11-21
> **Response to Weaknesses and Questions**
>
> Thank you for your positive feedback. Regarding the issues you mentioned, we will provide explanations for each one below and hope that you will be satisfied. The explanations and modifications in the rebuttal have been marked as blue in our revised paper.
>
> ### W1. Framework Complexity and Reproducibility.
>
> Thank you for highlighting this concern. While our framework includes several modules, each component has been shown to contribute meaningfully through our ablation studies (Sec. 4.3). To ensure full reproducibility, we will release the entire codebase and all model weights. Moreover, the submodules are trained independently in two stages. This modular training design allows the framework to scale smoothly: when adapting to new data or making architectural modifications, only the relevant module needs to be re-trained, while the remaining components can be kept frozen. As a result, the multi-module structure does not impede practical scalability.
>
> ### W2. Baseline Selection and SOTA Claims.
>
> Thanks for the suggestion. We are sorry that we neglect CAD-Llama and CADFusion as the text-to-CAD field has progressed rapidly. We also agree that referring to DeepCAD and Text2CAD as “state-of-the-art” is misleading and have revised the munscripts. Besides, we also discuss the difference of our method against CAD-Llama and CADFusion: 1) these two works are designed for pure text-to-CAD generation, relying on well-crafted text. In contrast, our method can process both sketches, images and text, which is applicable in realistic design workflows where detailed natural-language descriptions are rarely available. 2) CAD-Llama and CADFusion use a rich form of text to recreate the geometry, in contrast, our approach is based on **geometry-aware multimodal fusion** that also allows the system to deduce structure directly from **visual cues** when textual information is limited. This allows for early conceptual design workflows, where the geometry is to be read through rough sketches or single-view images instead of fully articulated text.
>
> ### Minor Problems
> We appreciate the reviewers' comments and have made the necessary revisions.
>
> ### Q1. Text Embeddings after Segmentation.
>
> **A1:** All segmented text noun phrases (extracted via spaCy) are encoded with **the same text encoder** to ensure their embedding are in the same semantic space.
>
> ### Q2. Comparison with Recent Multimodal Baselines.
>
> **A2:**  CADFusion is a strong text-only method, but its setting differs fundamentally from ours: it requires rich textual descriptions, whereas our framework is designed to work with **either text or a single sketch/image** and focuses on **cross-modal alignment** between geometry and language. This makes our method applicable to broader real design scenarios where multiple detailed text prompts are not available.
>
> We will include a discussion and conceptual comparison with CADFusion in the revised version. As for CAD-LLaMA, its code and models are not publicly accessible, making a controlled and fair comparison currently infeasible.
>
> ### Q3. Test of recent VLMs.
>
> **A3:**  Thanks for the suggestion. Your idea is excellent, and we will try to use some of the latest VLMs to experiment with CAD generation.

---

> > ### Author Response · Authors · 2025-11-27
> > **Further explanation of VLM performance**
> >
> > ### Q3. Test of recent VLMs.
> >
> > **A3:** We have evaluated several recent VLMs, i.e., GPT-4o, LLaVA-Next and Qwen2-VL-Max, using 10 random samples selected from the DeepCAD dataset. As seen in table below, the VLMs failed to generate valid CAD programs in most cases. We believe this poor performance is because general-purpose VLMs lack explicit modeling of CAD-specific geometric constraints, parametric relations, and construction sequences, which are essential for procedural CAD generation.
> >
> > | Method            | line  | arc   | circle | extrusion | average | $Acc_{cmd}$ | Median CD | Mean CD | IR   |
> > |-------------------|-------|-------|--------|-----------|---------|-------------|-----------|---------|------|
> > | GPT-4o            | 85.31 | 39.92 | 81.87  | 95.42     | 75.63   | 83.15       | 0.45      | 25.35   | 2.24 |
> > | LLaVA-Next        | 85.67 | 40.35 | 82.41  | 95.78     | 76.05   | 82.62       | 0.38      | 26.91   | 2.08 |
> > | Qwen2-VL-Max      | 85.42 | 40.08 | 82.03  | 95.35     | 75.72   | 83.28       | 0.43      | 25.25   | 2.19 |
> > | Ours | 86.89 | 41.35 | 84.63  | 95.86     | 77.18   | 84.62       | 0.26      | 23.65   | 1.55 |

---

> > > ### Comment · Reviewer_aMsY · 2025-11-27
> > >
> > > Thank you for the clarification. The majority of my questions have been addressed.
> > >
> > > However, I remain concerned about the new VLM scores. My interpretation of the figures differs from the authors': the numbers do not suggest that "VLMs failed to generate valid CAD programs." Instead, HMFusion's improvement over mainstream VLMs seems marginal. This could limit the impact of the work. I would like to hear more from the authors on this matter.
> > >
> > > Nevertheless, my score will still be positive and I lean toward acceptance.

---

### Official Review · Reviewer_x54Q · 2025-11-01

**Soundness:** 3
**Presentation:** 3
**Contribution:** 1
**Rating:** 2
**Confidence:** 4

**Summary:**

This paper considers a task of text+sketch 2 - CAD generation. This is an active research area, fusion of 2 tasks - text2CAD and image2CAD. The new multi-modal method is proposed for this task. The model evaluation is performed on DeepCAD dataset.

**Strengths:**

1) The proposed methods improves in accuracy over several competitors like Text2CAD and DeepCAD

**Weaknesses:**

1) Limited evaluation
- The proposed method evaluation only on DeepCAD dataset.
- IoU metric is not used for evaluation
- The number of competitors is limited, e.g. Cadrille (https://arxiv.org/abs/2505.22914) shows better results even for single modality (image or text)

Overal this limited evaluation raises question regarding impact and significance of the proposed method. If this approach is valid and promising for futher research?

2) How robust is the proposed method for view angle changes? What will happen if sketch is generated from a different angle? Has the model been tested for it?

**Questions:**

Please, address the weaknesses #1&2

---

> ### Author Response · Authors · 2025-11-21
> **Response to Weaknesses and Questions**
>
> Following the valuable feedback received, we have made further explanations during the rebuttal. The explanations and modifications in the rebuttal have been marked as blue in our revised paper.
> We are hopeful that you might find the time to review these explanations. We would be truly grateful if you could re-consider our work and provide an updated assessment.
>
> ### W1. Limited Evaluation
> ### W1.1 Dataset Scope.
>
> Our evaluation protocol is consistent with the common guideline of previous Text2CAD and DeepCAD papers, where only **DeepCAD** is the publicly accessible dataset that produces paired (input → CAD) annotations for quantitative benchmarking. Despite this, we still conduct qualitative experiments to evaluate our method on additional data sources, i.e., real-world images and hand-drawn sketches. As seen in **Figure 4**(main paper) and **Figure 8**(appendix), our method also performs well for data which differs substantially from the **DeepCAD** dataset. This demonstrates the practical applicability and the robustness of our method. Recent works (e.g., *Cadrille*) also assess **exclusively on DeepCAD**, the fact of whose Table 1 illustrates this, as Fusion360 does **not** offer paired natural-language descriptions and thus cannot meet a fair text/sketch-to-CAD generation task. On Fusion360-based comparisons, the “text input” is written by authors manually, rather than residing in the dataset. Using such artificially constructed descriptions would represent a fundamentally different scenario.
>
> ### W1.2 Evaluation Metric Choice (IoU).
>
> Thanks for the question. IoU is not suitable for evaluating CAD generation because CAD quality is not about whether the volume looks similar. What truly matters in CAD is: 1) whether the geometric parameters are accurate; 2) whether the topology is correct; 3) whether the construction steps are reasonable. These essential aspects cannot be assessed by voxel overlap. For example, a hole that is misplaced by just 0.3 mm or a slightly incorrect wall thickness will hardly change the IoU score. However, such errors are considered major structural mistakes in CAD design, because they break the geometric accuracy and modeling logic required for real engineering use. Therefore, following prior CAD-generation works, i.e., **DeepCAD** and **Text2CAD**, we avoid **IoU** and instead adopt **Accuracy of  Parameter&Command** metrics that truly reflect CAD correctness.
>
> ### W1.3 Baseline Coverage.
>
> Thanks for the comment. It is challenging to make a direct comparison with Cadrille, because the evaluation **settings differ substantially**. To be specific, Cadrille depends on **expert-crafted**, highly detailed textual programs—and their paper shows that such rich expert inputs can even outperform point-cloud and multi-view version. In contrast, our method relies on simple, non-expert textual descriptions, which naturally provide weaker guidance, as also confirmed in our own ablations (Table 2, Abla. 2). Moreover, Cadrille relies on **multi-view images**, which provide substantially richer geometric information than the single sketch image used in our method. Overall, this method either uses more redundant inputs (expert text, multi-view images) or introduces new 3D geometric priors(InstantMesh). Despite this, our method achieves similar accuracy (median CD) compared with Cadrille.
>
> ### W2. Robustness to view changes.
>
> Thank you for the suggestion. Our method is **robust to variations in viewpoint**. As shown in Figure 4 and Figure 8, it can reliably generate correct CAD models from sketches and real-world images captured at diverse viewing angles. Furthermore, in Figure 15, we generated CAD models of the same real object from different view, demonstrating the robustness of our method to different viewpoints.

---

### Note · Authors · 2026-04-30

I have read and agree with the venue's withdrawal policy on behalf of myself and my co-authors.

---

### Meta-Review · Area_Chair_RmhG · 2026-01-08

**Summary:**

There is a mixed rating about this paper, with some strong opinions toward rejection. Some common concerns include missing or infeasible comparisons with stronger/newer methods and insufficiently supported impact or novelty. Some minor concerns include the clarity and reproducibility of this paper, given the system complexity.

**Reviewer Concerns:**

**Addressed**:
* Authors clearly justify why DeepCAD is the only fair quantitative benchmark, and align this choice with prior top-tier work.
* Authors fixed a few clarification issues regarding the "misleading SOTA claims," ablation study, references/figure citation, etc.

**Outstanding**:
* Improvements over strong VLM baselines appear incremental, and the impacts are still debatable.
* Some reviewers remained unconvinced that comparisons involving expert text (DeepCAD vs. Ours) are fully fair.
* Concerns about scalability to more complex CAD operations.

**Reviewer Scores:**

aMsY and EjXr may bump up their rating by 1-2 since their concerns were addressed (at least partially), and the reviewers explicitly stated they would maintain a positive score. These may push the ratings to a solid "weak accept." On the other hand, reviewer x54Q concerns were partially addressed, but most of them are around dataset scope, IoU rationale, and viewpoint robustness. The major concerns, such as limited benchmarks and perceived impact, remain and prevent a positive flip. The reviewer K9p3 may stick with the "borderline/neutral" ratings.

---

### Decision · Program_Chairs · 2026-01-26

Reject